# A Preliminary Investigation about the Influence of WIMU PRO^TM^ Location on Heart Rate Accuracy: A Comparative Study in Cycle Ergometer

**DOI:** 10.3390/s24030988

**Published:** 2024-02-03

**Authors:** Joaquín Martín Marzano-Felisatti, Leonardo De Lucca, José Francisco Guzmán Luján, Jose Ignacio Priego-Quesada, José Pino-Ortega

**Affiliations:** 1Research Group in Sports Biomechanics (GIBD), Department of Physical Education and Sports, Faculty of Physical Activity and Sport Sciences, Universitat de València, 46010 Valencia, Spain; joaquin.marzano@uv.es; 2Laboratory of Human Performance Research, Centre of Health and Sport Sciences, University of Santa Catarina State, Florianópolis 88040-900, Brazil; leonardo_delucca@outlook.com; 3Research Group in Sports Technique and Tactics (GITTE), Department of Physical Education and Sports, Faculty of Physical Activity and Sport Sciences, Universitat de València, 46010 Valencia, Spain; 4Biovetmed & Sportsci Research Group, Department of Physical Activity and Sport, Faculty of Sport Sciences, University of Murcia, 30720 San Javier, Spain; josepinoortega@um.es

**Keywords:** technology, sports performance, internal load, wearables, sensor, precision

## Abstract

Technological development has boosted the use of multi-sensor devices to monitor athletes’ performance, but the location and connectivity between devices have been shown to affect data reliability. This preliminary study aimed to determine whether the placement of a multi-sensor device (WIMU PRO^TM^) could affect the heart rate signal reception (GARMIN^TM^ chest strap) and, therefore, data accuracy. Thirty-two physical education students (20 men and 12 women) performed 20 min of exercise in a cycle ergometer based on the warm-up of the Function Threshold Power 20 test in laboratory conditions, carrying two WIMU PRO^TM^ devices (Back: inter-scapula; Bicycle: bicycle’s handlebar—20 cm from the chest) and two GARMIN^TM^ chest straps. A one-dimensional statistical parametric mapping test found full agreement between the two situations (inter-scapula vs. bicycle’s handlebar). Excellent intra-class correlation values were obtained during the warm-up (ICC = 0.99, [1.00–1.00], *p* < 0.001), the time trial test (ICC = 0.99, [1.00–1.00], *p* < 0.001) and the cool-down (ICC = 0.99, [1.00–1.00], *p* < 0.001). The Bland–Altman plots confirmed the total agreement with a bias value of 0.00 ± 0.1 bpm. The interscapular back placement of the WIMU PRO^TM^ device does not affect heart rate measurement accuracy with a GARMIN^TM^ chest strap during cycling exercise in laboratory conditions.

## 1. Introduction

In elite sports, athletes and professional teams continuously search for performance improvement based on indicators that allow them to gain a competitive edge [1]. Among the wide variety of performance indicators, heart rate (HR) is commonly used to evaluate cardiac status [2,3]. Defined as a vital physiological variable, it helps in measuring internal load and prescribing exercise intensities [4,5]. Moreover, coaches need HR data in real-game contexts for decision-making [6]. This need has led to the development of devices that can provide live data during training and competition [3]. Among the different available options, chest straps have proven validity and reliability and have become the most widely used device for HR measurement [3,4,5].

Furthermore, it is essential to mention that sports performance monitoring should be more holistic and integrative than only measuring HR [7,8]. Thanks to the advances in sports technology, there is a tendency towards different device integration (e.g., Global Position Systems, accelerometers, gyroscopes, magnetometers, etc.) with the capacity for connectivity to external devices, providing more realistic data of athletes’ demands in real-game contexts [1,6,9]. In this sense, one of the most widely used devices by professional teams and researchers is the WIMU PRO^TM^, a validated wearable multi-sensor unit [10,11,12,13].

Regarding HR monitorisation, the WIMU PRO^TM^ includes a GARMIN^TM^ HR monitor [14,15]. Although the GARMIN^TM^ device is connected and coded via ANT+ technology to avoid interference, the question arises as to whether the placement of the GPS on the back could affect HR data accuracy [16]. In this sense, reference articles have shown that body mass index, device placement and device handling can be related to HR measurement errors [17]. For example, wearable photoplethysmography technique-based sensors can create a significant variation in heart rate outcomes based on the place (i.e., proximal vs. distal; dominant vs. non-dominant) applied, even though the same device is used [18]. Moreover, the back placement of the multi-sensor is widely adopted in team sports [19,20] and is now gaining popularity in individual sports [21,22] and various disciplines [23]. Therefore, the authors find it relevant to assess whether the back placement of the sensor has any impact on the accuracy of HR data collected by chest straps during physical activities.

For the above reasons, this preliminary research aims to determine whether the multi-sensor device’s placement could affect the HR signal reception and, thus, the accuracy of the data in athlete monitorisation during a controlled laboratory situation. It was hypothesised that the placement of the WIMU PRO^TM^ multi-sensor device on the back may affect the HR signal reception and, thus, the data accuracy.

## 2. Materials and Methods

### 2.1. Experimental Approach to the Problem

A cycling exercise test based on the warm-up of the Function Threshold Power 20 (FTP20) test was performed in the laboratory [24] using two WIMU PRO^TM^ (RealTrack System, Almeria, Spain) devices to verify whether there is a difference in HR values when placing the tracking device in different locations (scapula height and bicycle’s handlebar—20 cm from the chest) (Figure 1). Raw data were extracted from both devices to analyse intra-class correlations, biases and significant differences between them.

### 2.2. Participants

Thirty-two physical education students (20 men and 12 women) enrolled at the University of Murcia in the 2023/2024 academic period volunteered to participate in the study (Men: 20 ± 0 years; 175.5 ± 4.4 cm; 78.9 ± 10.4 kg—Women: 20 ± 0 years; 163.5 ± 4.8 cm; 58.3 ± 6.4 kg). All participants were free from injury during testing and were asked to maintain their regular hydration and feeding before the test. The study was approved by the local Ethics Committee (ID: 3495/2021) and followed the ethical recommendations for studies in humans established by the Declaration of Helsinki.

### 2.3. Procedure

Participants were asked to refrain from performing vigorous-intensity exercise the day before each session while maintaining a similar diet throughout the testing. They arrived at the laboratory 30 min before tests to be instructed about test procedures. The volunteers wore two HR monitor chest belts (GARMIN^TM^, Garmin Ltd., Olathe, KS, USA) attached to the ribcage under the musculus pectoralis major (Figure 1). Each HR monitor was synchronised with WIMU PRO^TM^ devices by ANT+ technology, which collects data at 4 Hz. The WIMU PRO^TM^ devices were placed in two locations: (1) scapula height and (2) cycle ergometer’s handlebar—20 cm from the chest—for further comparison (Figure 1). The cycling test was conducted on a cycle ergometer (BODYTONE EX4, Bodytone International Sports S.L., Murcia, Spain) based on the warm-up of the 20 min trial (FTP20) to analyse possible differences between HR data in both conditions. Participants performed a cycling test consisting of a warm-up (5 min of light pedalling at ~100 W, followed by three 1 min efforts pedalling at 100 rpm interspersed with a 1 min recovery of light intensity cycling), a 5 min all-out effort and, finally, 4 min of light pedalling as a cool-down (Figure 2). A specific pacing tactic was not suggested, although participants were encouraged to achieve the highest mean power during the 5 min all-out effort. During the test, participants were blinded to intensity but were allowed to see time and cadence for individual pacing strategies. WIMU PRO™ software (SPRO™, version 989, Realtrack Systems SL, Almería, Spain) was used to compute HR. The WIMU PRO^TM^ is an inertial recording device that monitors physical activity and movement in real time. This inertial device uses different sensors to record the data (GNSS, UWB, 4 × 3D accelerometer, 3 × 3D gyroscope, 3D magnetometer, barometer, WIFI, bluetooth, ANT + and USB connection). The information recorded by the sensors was converted into quantitative data using the software SPRO^TM^. Raw data were exported to an Excel Spreadsheet (Microsoft Corporation, Redmond, WA, USA) for further analysis.

### 2.4. Statistical Analysis

Data are presented as mean, standard deviation (SD) and Confidence Interval at 95% (95%CI). Firstly, to compare the differences between locations of both WIMU PRO^TM^ devices in HR, one-dimensional statistical parametric mapping (SPM) techniques were implemented to analyse a time-series signal along the test [25]. For this analysis, data were normalised considering the test duration from 0% to 100%. The SPM1D Python (Anaconda Navigator 2.3.2) was used by applying a paired t-test (as the data presented a normal distribution). Secondly, the intra-class correlation coefficients (ICCs) of HR between the locations of both WIMU PRO^TM^ devices were obtained using RStudio (version 2023.06.0, package “irr”), based on a single-rater measurement, absolute agreement and a 2-way random-effects model. ICC values were classified as 1.00–0.81 (excellent), 0.80–0.61 (very good), 0.60–0.41 (good), 0.40–0.21 (reasonable) and 0.20–0.00 (deficient) [26]. Finally, Bland–Altman plots were used to examine the agreement between HR derived from both devices [27].

## 3. Results

Figure 3 shows the HR raw data obtained during the cycling test in both sensor positions. The mean value obtained for both sensors was 117.5 ± 8.0 bpm for the warm-up, 164.7 ± 13.2 bpm for the 5 min all-out effort, and 147.4 ± 10.7 bpm for the cool-down.

SPM analysis did not present differences between sensor positions at any moment of the test (Figure 4, *p* > 0.05). The left panel of the figure shows how the heart rate of both sensors overlaps throughout the test, and the right panel shows the statistics of the paired t-test, which has a value of 0 throughout the test, not being altered at any moment.

Mean and SD values were obtained for each phase (warm-up, 5 min all-out effort and cool-down), and ICC values were calculated (Table 1). ICC values were excellent between the locations of both devices, obtaining an ICC value of 1.00 (*p* < 0.001) in the three phases (warm-up, 20 min time trial and recovery).

Bland–Altman plots (Figure 5) show that the bias between devices in the three phases was 0.0 ± 0.1 bpm, with no modification of this bias depending on the mean HR. The ±1.96 standard deviations of the bias (which is the same as the 95%CI of the bias) for the warm-up, 5 min all-out effort and cool-down were ±0.2 bpm, ±0.3 bpm and ±0.2 bpm, respectively.

## 4. Discussion and Conclusions

This preliminary research aimed to determine if the WIMU PRO^TM^ placement could affect the HR signal reception and, thus, the data accuracy in athletes’ monitorisation during controlled laboratory situations. The main results were that no differences were observed between the two device locations assessed, showing an excellent intra-class correlation coefficient at any moment of the cycling test and a bias value between both locations of 0.00 ± 0.1 bpm. The authors’ hypothesis was rejected as the WIMU PRO^TM^ on the back did not affect the HR signal reception and data accuracy.

The statistical analysis demonstrated the total agreement between the two situations (inter-scapula vs. bicycle’s handlebar) without any influence of intrinsic or extrinsic factors. Although other HR measurement systems are affected by position or athlete characteristics, the SPM test reveals no differences during the entire trial, regardless of exercise intensity and athletes’ individuality. In addition, excellent intra-class correlation values were found during the warm-up (ICC = 0.99, [1.00–1.00], *p* < 0.001), the 5 min all-out effort (ICC = 0.99, [1.00–1.00], *p* < 0.001) and the cool-down (ICC = 0.99, [1.00–1.00], *p* < 0.001). The Bland–Altman plots also confirm these results with mean bias values of 0.00 and minimum standard deviations with upper and lower limits close to zero.

Research on the impact of device placement on heart rate measurements has yielded mixed results. Jung et al. [28] found that the placement of a Fitbit HR did not significantly influence heart rate measurements, while Brage et al. [29] reported that the placement of the Actiheart at the level below the sternum yielded cleaner heart rate data. Our results align with previous studies that aimed to verify HR measurement accuracy in exercise and sports contexts [3,28,30]. Specifically, the WIMU PRO^TM^ devices focused on the importance of multi-device integration for a more comprehensive understanding of athletes’ physical demands (external and internal load management) [1,12]. Moreover, the results contribute to previous studies that delved into how device location could affect data accuracy [12]. Although WIMU PRO^TM^ positions have been shown to affect external load data [31], HR measurements through GARMIN^TM^ chest straps have been proven to be accurate, independent of the WIMU PRO^TM^ location [12]. A possible explanation for our findings is the high accuracy of chest strap monitors, with better agreements with ECG measurements than other technologies. Specifically, the GARMIN^TM^ chest strap telemetry for heart rate works using a piezoelectric transducer to detect heart sounds on the chest surface [32]. Parak et al. [30] found that chest straps are more accurate than vests, with a mean absolute percentage error (MAPE) of 0.76% for chest straps and 3.32% for vests. Pasadyn et al. [33] also found that chest strap monitors, such as the Polar H7, had the greatest agreement with ECG measurements among others, followed by the Apple Watch III. In fact, the chest strap can more easily be adjusted tightly to different chest circumferences compared to other non-invasive devices, making it suitable for continuous monitoring during various types of exercise. Furthermore, heart rate monitors that use ANT+ technology have been designed with better signal conditioning to prevent misleading displays, enhancing accuracy and usability [34,35].

Furthermore, considering that most team sports use these multi-device systems to monitor athletes’ internal and external load, this research confirms the HR accuracy with the most extended device locations (HR chest strap and back interscapular GPS). Coaches, technical staff and athletes can be confident with the HR data obtained through the system proposed in this research.

This study was considered to be preliminary due to its limitations and limited scope. Firstly, the HR recordings were not compared with a more rigorous approach, such as an electrocardiogram. Secondly, the results apply only to the HR monitor device tested. Thirdly, the devices and placements were tested on young, healthy volunteers during cycling in a laboratory setting. Indoor cycling may not fully represent young adults’ physical activity. Hence, generalisations cannot be made for other sports and exercise environments, children or older adult age groups, or individuals of different body sizes or clinical populations. Future studies could analyse various types of fitness trackers and a broad range of activities ranging from sedentary to vigorous intensities with balanced maintenance in each intensity.

In conclusion, the interscapular back placement of the WIMU PRO^TM^ device does not affect HR accuracy measurement with a GARMIN^TM^ chest strap during cycling exercise in laboratory conditions. These results confirm the reliability of multi-sensor devices in sports performance monitoring, especially underlining the consistency of HR data considering device location. This study supports multi-sensor wearable technology in professional sports contexts and gives confidence to coaches and athletes in integrating these metrics for performance analysis and decision-making. Future research should aim to replicate these findings in different sports environments, with different multi-sensor devices and populations, to ascertain the study findings.

## Figures and Tables

**Figure 1 sensors-24-00988-f001:**
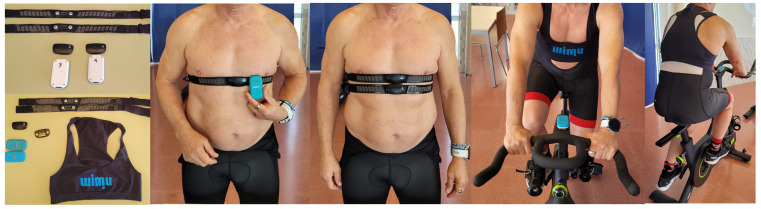
Equipment and participant instrumentation during testing.

**Figure 2 sensors-24-00988-f002:**
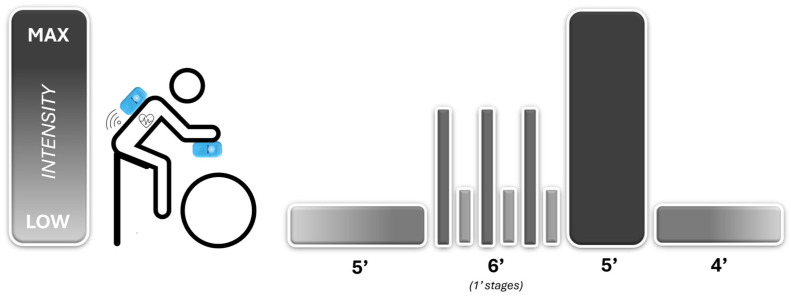
Study design, device placement and intensities performed by participants.

**Figure 3 sensors-24-00988-f003:**
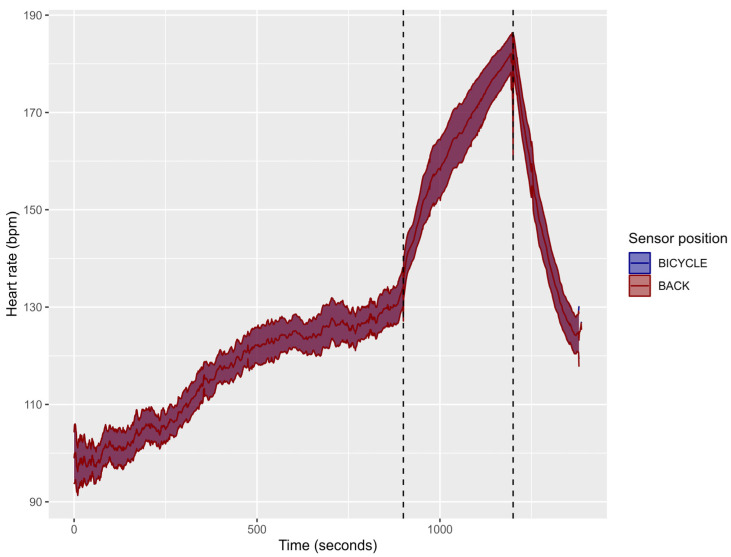
Mean (line) and Confidence Interval at 95% (shaded interval) of the heart rate response during the cycling warm-up (data left of the first vertical dotted line), 5 min all-out effort (data between the two vertical dotted lines) and cool-down (data right of the second vertical dotted line), with the WIMU PRO^TM^ devices placed in two different locations (Back: scapula height; Bicycle: bicycle’s handlebar—20 cm from the chest). The colour of the signal presented is the result of the two colours of both sensors being overlapped.

**Figure 4 sensors-24-00988-f004:**
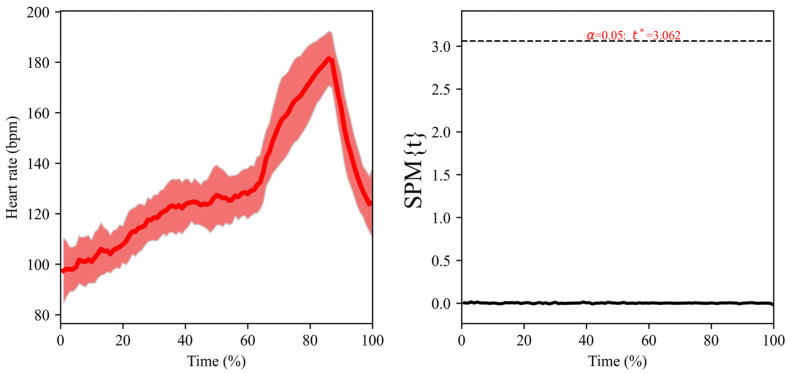
(**Left**): mean (line) and standard deviation (shaded interval) of the heart rate response during all the cycling tests (warm-up, a 5 min all-out effort and cool-down), with the time normalised as a percentage of the total time of the test. Data from the two locations of the WIMU PRO^TM^ devices are shown superimposed. (**Right**): the paired *t*-test (SPM1D) compared the heart rate between both locations of WIMU PRO^TM^ devices (scapula height vs. bicycle’s handlebar—20 cm from the chest). The *y*-axis displays the one-dimensional SPM {t}. A significant effect (*p* < 0.05) is present where the black line is above the upper horizontal dotted line.

**Figure 5 sensors-24-00988-f005:**
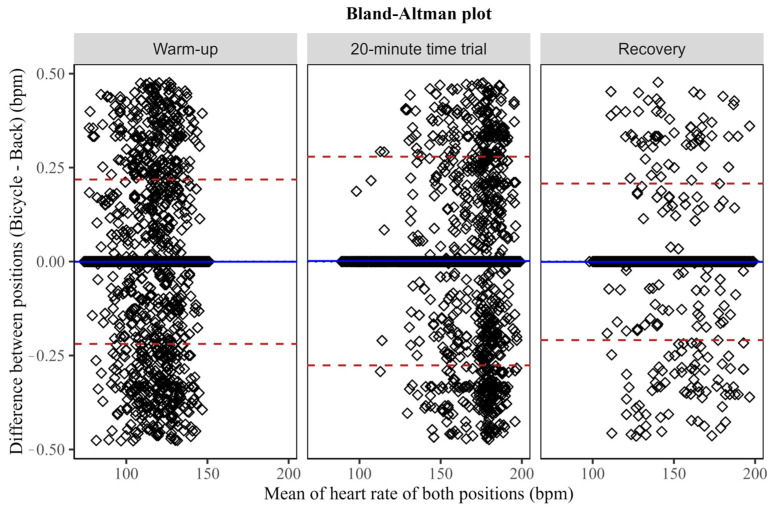
Bland–Altman plots for heart rate measurement between the WIMU PRO^TM^ devices placed in two different locations (Back: scapula height; Bicycle: bicycle’s handlebar—20 cm from the chest). The central continuous blue line represents the absolute average difference between devices (bias), and the upper and lower red lines represent ±1.96 standard deviations.

**Table 1 sensors-24-00988-t001:** Mean and standard deviation (SD) for the WIMU PRO^TM^ devices placed in two different locations (Back: scapula height; Bicycle: bicycle’s handlebar—20 cm from the chest) for each phase of the test (warm-up, 5 min all-out effort and cool-down). Intra-class correlation coefficient (ICC) between both locations was determined with its Confidence Interval at 95% (95%CI).

Bicycle			Back		ICC Results			
Phase	Mean	SD	Mean	SD	ICC	*p*-Value	95CI Lower	95CI Upper
Warm-up	117.5	8.0	117.5	8.0	1.00	<0.001	1.00	1.00
5 min all-out effort	164.7	13.2	164.7	13.2	1.00	<0.001	1.00	1.00
Cool-down	147.4	10.7	147.4	10.7	1.00	<0.001	1.00	1.00

## Data Availability

Data are contained within the article.

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
