# Peer review of "A Preliminary Investigation about the Influence of WIMU PROTM Location on Heart Rate Accuracy: A Comparative Study in Cycle Ergometer"

_sensors, 2024, doi:10.3390/s24030988_

Round 1

Reviewer 1 Report

Comments and Suggestions for Authors

Review for Sensors

The manuscript presents an interesting work about monitoring athlete's performance using multi-sensor devices.

The article is well-structured and written.

However, there are some suggestions to be included in the manuscript to add missing data or make it easier for the reader.

Page 2, sections 2.1 and 2.2 start with the same number of students. As your section 2.2 is about participants, it is better to be given here.

There are missing pictures of the participants during the testing procedure.

There are missing pictures of at least one participant wearing the HR monitor chest belts and multi-sensory devices.

All the figures have low resolution.

Section 4, is suggested to include conclusions so should be Discussions and conclusions. Or you can add another section about conclusions.

The description of each figure should be shortened and the explanation should be given in the text. For example: Figure 3 is given: “The central continuous blue line represents the absolute average difference between devices (bias), and the upper and lower red lines represent ± 1.96 standard deviations.” it is a description.

Reviewer 2 Report

Comments and Suggestions for Authors

Review for the paper title: Influence of WIMU PROTM Location on Heart Rate Accuracy: A Comparative Study in Cycle Ergometer

1.      Literature review is not sufficient, more literature review is requited for the paper.

2.      Methodology section i.e. (procedure) is very brief and more details are needed to be added such as type of diet the participants used, how you know they follow the same diet across the whole period of time, why they need to maintain same diet and what is the impact of this. You need to describe all used devices in terms of specifications. You need also to add a block diagram to summaries all stages of the methodology used. In addition, you need to add some Figures for the used devices or the whole experiment. All written methodology needs to be revised to allow repeatability and usability of the proposed scientific research.

3.       Figure 1 needs to be regenerated as there is only one color appears on the figure and the legend define two colours. In addition, I can see that the two colors overlapped with each other and generated a new color which is not defined within the legend of the graph and this is misleading the reader.

4.      More discussion of results is needed after each graph and comparison between males and females is needed, differences between the results obtained from differences between participants need to be highlighted and discussed (if there are any).

5.      The presented results are very limited.

 All the best 

Reviewer 3 Report

Comments and Suggestions for Authors

My main concerns about this paper are about the scope and contribution of this work. The study's scope is limited to the WIMU PRO™ and two body positions. A broader comparison across different commercial products and various body-sensing positions would enhance the paper's value. It raises the question of whether different products perform differently in various body locations. Popular sensing positions like the wrist (for smartwatches and bands) and the ear (for earpieces) are omitted, though it may not apply to the current focused product.

Another issue is benchmarking for the experiments. The authors could consider using ECG to validate heart rate data accuracy. Comparing results with other products like smartwatches would also be beneficial.

The paper is confined to a laboratory setting and cycling, which may not accurately represent other sports or real-world conditions. The participant demographic, predominantly young, physically fit education students, might also constrain the broader application of the findings to varied populations and age groups.

Furthermore, the paper's figures could benefit from enhanced resolution, as the current version seems to be blurred.

Round 2

Reviewer 2 Report

Comments and Suggestions for Authors

Thanks for considering all the comments.  

Author Response

We would thank the reviewer for the previous comments that helped to improve our paper.

Reviewer 3 Report

Comments and Suggestions for Authors

I am pleased to see the enhanced resolution of the figures in this paper, as it improves their clarity. However, I noticed that while the authors acknowledge certain limitations in the conclusion and modify the scope of their claims, there has been limited effort to conduct additional experiments. Acknowledging limitations is important but addressing them more directly could help enhance the paper's quality. To this end, I suggest the inclusion of more subjects, with a focus on diverse participant demographics, testing the devices on various human body positions, and/or conducting a more comprehensive analysis of current data. This would not only strengthen the validity of the findings but also broaden the paper's scope and impact.

Author Response

Dear reviewer, we completely understand your comments and we have considered them. Although we agree with you that these additional analyses would improve considerably the scope of our paper, we also consider that this needs time (to recruit the participants, analyze the data, etc.) and it is not possible to do in the time that we have for the review (10 days). Moreover, we consider that this new analysis would transform our manuscript into another one, as it will be necessary to modify considerably all the sections. For these reasons, and because we agree with your comments, we have considered our work as preliminary, including this concept in the title, objectives and discussion of the limitations of the work.